# FraIR: Fourier Recomposition Adapter for Image Restoration

## Abstract

Restoring high-quality images from degraded inputs is a core challenge in computer vision, especially under diverse or compound distortions. While large-scale all-in-one models offer strong performance, they are computationally expensive and poorly generalize to unseen degradations. Parameter-Efficient Transfer Learning (PETL) provides a scalable alternative, but most methods operate in the spatial domain and struggle to adapt to frequency-sensitive artifacts like blur, noise, or compression. We propose **FraIR**, a Fourier-based Recomposition Adapter for image restoration that enables efficient and expressive adaptation in the spectral domain. FraIR applies a 1D Fourier Transform to decompose token features into frequency components, performs low-rank adaptation via spectral projections with learnable reweighting, and reconstructs the adapted signal using an inverse transform gated by task-specific modulation. Integrated as plug-and-play modules within Transformer layers, FraIR is reparameterizable for zero-latency inference and requires less than 0.5% additional parameters. Extensive experiments across denoising, deraining, super-resolution, and hybrid-degradation benchmarks show that FraIR outperforms prior PETL methods and matches or exceeds fully fine-tuned baselines demonstrating strong generalization with minimal cost. Unlike prior Fourier-based approaches that focus on generative modeling or static modulation, FraIR offers dynamic, degradation-aware recomposition in frequency space for efficient restoration.

## 1 Introduction

Image restoration in the wild is fundamentally hard. Real-world degradations encompassing blur, noise, compression, rain, low-light are not only diverse but compound, and their effects are often most distinguishable in the frequency domain. High-frequency components encode critical details like textures and edges, while low frequencies govern overall structure. Yet, despite this, most restoration models operate exclusively in the spatial domain, ignoring a key axis of signal separability.

At the same time, real-world deployment imposes another strict constraint: efficiency. While large, all-in-one restoration models Li et al. (2022b); Zamfir et al. (2024); Cui et al. (2025) can be trained to simultaneously handle multiple degradations, they are computationally expensive and brittle, thus incapable of generalizing beyond what they've seen. Fine-tuning them per task is prohibitively costly. This presents a unique opportunity: *can we build degradation-aware, frequency-sensitive, parameter-efficient modules that adapt large pre-trained models to restoration tasks without sacrificing quality or generalization?*

We begin by observing that different degradations manifest as distinct spectral distortions. To visualize this, we compute log-ratio maps between the Fourier spectra of degraded and ground truth images as shown in Figure 1. These maps highlight how each degradation uniquely amplifies or suppresses specific frequency bands — blur attenuates high frequencies, noise spikes mid-high bands, compression introduces structured band gaps, and so on. Such clear task-specific frequency signatures motivate a frequency-aware strategy for adaptation.

We introduce FraIR (**F**ourier **R**ecomposition **A**dapter for **I**mage **R**estoration), a novel adapter design that brings frequency-domain adaptation to parameter-efficient transfer learning (PETL). Instead of adding spatial-domain prompts Potlapalli et al. (2024) or bottlenecks Li et al. (2022a), FraIR operates in the Fourier domain: it decomposes token embeddings via a 1D Discrete Fourier Transform, applies

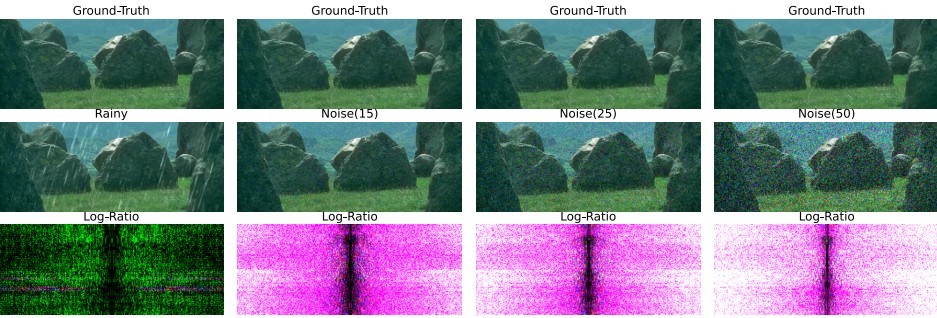

Figure 1: Fourier Log-ratio maps in the Fourier domain highlight how different degradations uniquely alter frequency components — pink indicates amplification, green represents attenuation. These structured spectral shifts motivate the need for frequency-aware, degradation-specific adaptation as enabled by FraIR.

low-rank spectral projections with learnable reweighting, and recomposes the features using an inverse transform modulated by degradation-aware gating. This simple yet powerful mechanism allows FraIR to target frequency bands selectively amplifying high-frequency detail where needed, or suppressing noise in low-frequencies without touching the backbone weights. FraIR thus represents a shift in PETL for image restoration, from spatial to spectral adaptation, from uniform task-agnostic modules to degradation-aware frequency recomposition. By leveraging frequency-domain characteristics of corruption, FraIR provides an efficient mechanism for selectively enhancing or suppressing specific bands without modifying backbone weights. This design opens a path toward scalable, adaptable restoration systems that generalize across unseen degradations while remaining lightweight and easy to deploy.

- We introduce **FraIR**, a novel frequency-domain adapter architecture for parameter-efficient image restoration. Unlike prior PETL methods that operate in the spatial domain, FraIR performs adaptation via low-rank projections in the complex Fourier space, enabling degradation-aware modulation along frequency components.
- We formulate a spectral recomposition mechanism based on structured low-rank decomposition and learnable spectral reweighting. This allows efficient control over the adapter's expressive capacity through SVD rank and supports exact reparameterization into the backbone weights for zero-latency inference.
- We conduct extensive controlled comparisons against spatial, wavelet, and low-rank adapter variants, validating FraIR's superiority across multiple restoration tasks and backbones. Our method achieves state-of-the-art adaptation performance while updating less than 0.5% of model parameters.

## 2 RELATED WORK

**Image Restoration.** Image restoration has progressed from conventional algorithms to deep learning-based models, with CNNs Zhang et al. (2018c); Zamir et al. (2021); Zhang et al. (2017b); Ren et al. (2018) and Transformers Liang et al. (2021); Zamir et al. (2022); Liu et al. (2024); Chen et al. (2022b); Korkmaz & Tekalp (2024) achieving strong performance in tasks such as denoising Zhang et al. (2017b); Chen et al. (2022b), dehazing Ren et al. (2020); Lu et al. (2024); Zhang et al. (2024), and deraining Jiang et al. (2020); Yan et al. (2024); Chen et al. (2024a;b). UNet-style encoder-decoder architectures and skip connections are commonly used, often enhanced with attention modules. CNNs provide computational efficiency, while Transformers offer superior global modeling at higher cost. Hybrid models, such as Restormer Zamir et al. (2022), aim to balance these trade-offs using linear-complexity attention and gated feed-forward designs. While most works focus on single degradation types, multi-degradation restoration remains underexplored. Some approaches target weather-related corruptions using parallel branches or contrastive learning, but typically require prior knowledge of the degradation, limiting their generalizability. All-in-one restoration models like AirNet Li et al. (2022a), DaAIR Zamfir et al. (2024) and AdaIR Cui et al. (2025) aim to generalize across multiple

degradations. AirNet uses contrastive degradation encoding, DaAIR employs a degradation-aware learner to jointly model shared and distinct degradation features, and AdaIR extracts low- and high-frequency information from the input features, guided by the adaptively decoupled spectra of the degraded image. In contrast to these methods, we do not rely on explicit degradation modeling or large all-in-one networks. Instead, we introduce a lightweight, frequency-domain adapter that enables scalable, degradation-aware adaptation via spectral recomposition achieving strong performance with minimal additional parameters.

**Parameter-Efficient Transfer Learning.** The growing size of deep networks has motivated Parameter-Efficient Transfer Learning (PETL) methods that adapt pre-trained models using a small number of additional parameters. In vision tasks, this includes approaches such as prompt tuning Jia et al. (2022a); Potlapalli et al. (2023), adapters Houlsby et al. (2019b), LoRA Hu et al. (2022), and scaling-and-shifting strategies Lian et al. (2022). Visual Prompt Tuning (VPT) introduces learnable tokens prepended to input sequences, while Adapter modules insert lightweight bottlenecks after attention or MLP layers. LoRA adds low-rank updates to specific weight matrices, and more recent methods like AdaptIR Guo et al. (2024) use Mixture-of-Experts structures to model degradation diversity. While effective in classification or segmentation, these spatial-domain PETL approaches often struggle in image restoration, where degradations exhibit distinct spectral characteristics. Moreover, existing methods frequently produce homogeneous representations across tasks, limiting their generalization capacity. In light of this, our method performs adaptation directly in the frequency domain via spectral decomposition and reweighted low-rank projections. This enables degradation-specific modulation while preserving PETL's efficiency thus achieving superior generalization across diverse and unseen corruptions with minimal parameter overhead.

**Fourier Transform for Image Restoration.** Fourier-based techniques Gao et al. (2024); Borse et al. (2024); Zeng et al. (2024); Ly & Nguyen (2025) have gained traction in image restoration for their ability to capture global context and frequency-localized degradations. Early works such as FDRNet Li et al. (2020) and FCA-Net Qin et al. (2021) integrate frequency priors into CNNs, while AirNet Li et al. (2022b) uses Fourier contrastive learning to guide degradation-aware restoration. More recent models like FSRNet Yu et al. (2023) dynamically adapt frequency features for improved generalization. SVDiff Han et al. (2023) applies spectral-domain singular value decomposition to diffusion models, enabling low-rank adaptation in generative settings. FSRNet and SVDiff, which focus on spectral compression or layer-wise modulation, on the contrary, our method introduces a unified framework that integrates spectral decomposition, low-rank recomposition, and dynamic reweighting within a plug-and-play adapter. FourierFT Gao et al. (2024) learns adaptation in the frequency domain by projecting the update matrix $\Delta W$ into the Fourier basis and optimizing only a small subset of spectral coefficients, then reconstructing the full update via an inverse DFT. This yields parameter savings by operating on compressed frequencies. FouRA Borse et al. (2024) applies low-rank adaptation directly in Fourier space. It transforms features to the frequency domain, learns low-rank up/down projections, and uses an input-dependent gating mask inside the low-rank subspace to vary the effective rank, while the underlying Fourier transform (spectral basis) is fixed and shared across inputs. Unlike FourierFT and FouRA, FraIR introduces two mechanisms tailored for image restoration: **(i) a degradation-aware channel-wise gate** $\mathbf{g} \in (0,1)^D$, which modulates spectral channels and controls the strength of the frequency-domain update, and **(ii) spectral recomposition** through a shared low-rank basis defined by $U$, which flexibly fuses the adapted frequency components back into the feature space. This allows a single adapter to generalize across diverse degradations without architectural changes or generative supervision.

## 3 FRA: FOURIER RECOMPOSITION ADAPTER

**Adapter Recomposition.** Adapter Recomposition extends traditional adapter-based fine-tuning by decomposing the update matrix into multiple low-rank components and dynamically reassembling them based on task-specific relevance. Let a pre-trained layer be parameterized by a weight matrix $W_0 \in \mathbb{R}^{k_2 \times k_1}$, mapping an input vector $z_{\text{in}} \in \mathbb{R}^{k_1}$ to an output vector $z_{\text{out}} \in \mathbb{R}^{k_2}$. Each low-rank subspace is defined by matrices

$$A_i \in \mathbb{R}^{k_2 \times r_i}, \qquad B_i \in \mathbb{R}^{r_i \times k_1}, \tag{1}$$

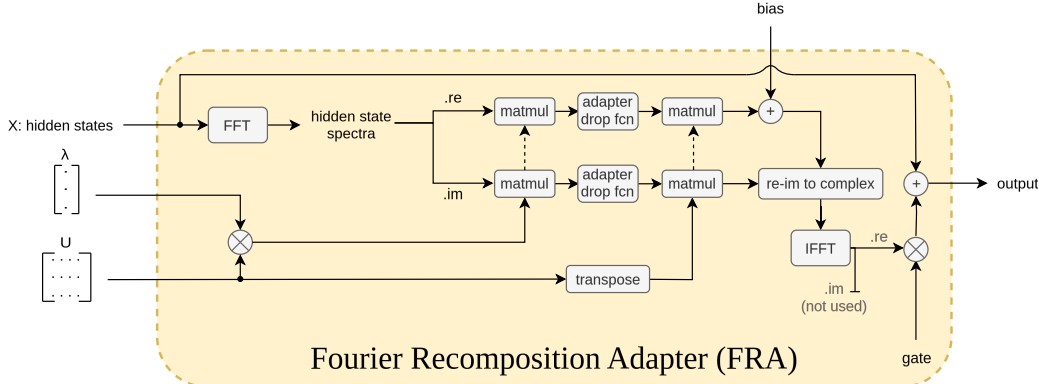

Figure 2: Architecture of the proposed Fourier Recomposition Adapter (FRA). Input features are transformed to the frequency domain via 1D FFT. Real and imaginary components are separately adapted using low-rank projections, followed by inverse FFT to reconstruct the output. A gated residual connection integrates the adapted frequency features into the original hidden representation.

where $r_i$ is the rank of the $i$-th component. The product $A_i B_i$ therefore has shape $k_2 \times k_1$, matching the dimensionality of $W_0$.

We combine these components using learned scalar coefficients $\gamma_i \in \mathbb{R}$. The resulting adapter update:

$$\Delta W_{\text{adapter}} = \sum_i \gamma_i A_i B_i. \tag{2}$$

The adapted layer output is then computed as $z_{\text{out}} = W_0 z_{\text{in}} + \alpha(\sum_i \gamma_i A_i B_i) z_{\text{in}}$. where $\alpha \in \mathbb{R}$ is a learnable scaling factor controlling the strength of the adaptation. This formulation enables parameter-efficient updates while allowing the model to dynamically emphasize task-relevant low-rank subspaces.

### 3.1 FOURIER-BASED RECOMPOSITION ADAPTER

Our goal is to perform parameter-efficient adaptation directly in the frequency domain, where many degradations (e.g., blur, noise, ringing) exhibit structured and separable behavior. This motivates an adapter that modulates frequency components rather than spatial activations.

**Step 1: Fourier projection.** Given a feature map $X \in \mathbb{R}^{N \times D}$, where $N$ is the sequence length and $D$ the channel dimension, we first project features into the complex Fourier basis:

$$\hat{X} = \mathcal{F}(X), \qquad \hat{X} \in \mathbb{C}^{N \times D}. \tag{3}$$

**Step 2: Low-rank spectral transformation.** Instead of applying a full-rank weight update, we introduce a compact low-rank operator in the Fourier space. Let $U \in \mathbb{R}^{D \times D'}$ be a learnable projection with $D' \ll D$, and let $\Lambda \in \mathbb{R}^{D \times D'}$ denote a learnable scaling matrix that is applied elementwise to $U$. We define

$$\hat{Z} = \hat{X} (U \odot \Lambda), \tag{4}$$

where $\odot$ denotes elementwise multiplication and thus $\hat{Z} \in \mathbb{C}^{N \times D'}$.

**Step 3: Spectral recomposition.** We reconstruct the adapted frequency representation via

$$\hat{Z}' = \hat{Z} U^\top + \mathbf{b}, \qquad \mathbf{b} \in \mathbb{R}^D, \tag{5}$$

where $\mathbf{b}$ is a learnable channel-wise bias that is broadcast across tokens.

**Step 4: Inverse Fourier mapping.** The adapted representation is mapped back to the spatial domain using the inverse DFT:

$$\tilde{X} = \mathcal{F}^{-1}(\hat{Z}'). \tag{6}$$

**Step 5: Residual gating.** Finally, we blend the original and adapted features through a learnable channel-wise gate:

$$X_{\text{out}} = X + \mathbf{g} \odot \tilde{X}, \qquad \mathbf{g} \in \mathbb{R}^D, \tag{7}$$

which enables channel-dependent control of the Fourier-domain update strength.

## 3.2 Integration into Transformer Layers

The FraIR adapter can be seamlessly inserted into Transformer architectures (e.g. Liang et al. (2021)), positioned before both the Multi-Head Attention (MHA) and Feed-Forward Network (FFN) modules. Let $X$ denote the input to a Transformer block; the adapted forward pass becomes:

$$X' = \text{MHA}(\text{FraIR}(\text{LN}(X))) + X, \tag{8}$$

$$X_{\text{out}} = \text{FFN}(\text{FraIR}(\text{LN}(X'))) + X'. \tag{9}$$

where, LN denotes layer normalisation, and FraIR modules before MHA and FFN are independent and maintain separate projection parameters.

**Reparameterization for inference.** Since the proposed FRA module is linear with respect to the input features, its effect can be merged into the first linear layer of the FFN at inference time. Let the original FFN be

$$Y = \text{GELU}(X'W_1 + \mathbf{b}_1)W_2 + \mathbf{b}_2, \tag{10}$$

where $X' \in \mathbb{R}^{N \times D}$, $W_1 \in \mathbb{R}^{D \times D_{\text{ff}}}$, $W_2 \in \mathbb{R}^{D_{\text{ff}} \times D}$, and $\mathbf{b}_1, \mathbf{b}_2$ are the FFN biases. With FRA applied before the FFN, we obtain

$$Y = \text{GELU}(\text{FRA}(X')W_1 + \mathbf{b}_1)W_2 + \mathbf{b}_2. \tag{11}$$

Because FRA is a linear operator in the feature space, there exist an equivalent matrix $M \in \mathbb{R}^{D \times D}$ and a channel-wise bias $\mathbf{c} \in \mathbb{R}^D$ such that

$$\text{FRA}(X') = X'M + \mathbf{1}\mathbf{c}^\top, \tag{12}$$

where $\mathbf{1} \in \mathbb{R}^N$ is an all-ones vector that broadcasts the bias across tokens. Folding the FRA operator into the FFN yields

$$Y = \text{GELU}(X'W_1' + \mathbf{b}_1')W_2 + \mathbf{b}_2, \tag{13}$$

with the reparameterized weights

$$W_1' = MW_1, \qquad \mathbf{b}_1' = \mathbf{c}^\top W_1 + \mathbf{b}_1. \tag{14}$$

Thus, all FRA parameters $(U, \Lambda, \mathbf{b}, \mathbf{g})$ can be fused into a single reparameterized FFN layer during inference, incurring no additional computational cost.

## 4 Experiments

### 4.1 Experimental Settings

**Datasets.** For color image denoising, we combine images of BSD400 Arbelaez et al. (2010), and WED Ma et al. (2016), datasets for model training; the BSD400 contains 400 training images, while the WED dataset consists of 4,744 images. Starting from these clean images of BSD400 Arbelaez et al. (2010), and WED Ma et al. (2016), we generate their corresponding noisy versions by adding Gaussian noise with varying levels $\sigma \in 15, 25, 50$. We evaluated denoising results on BSD68 Martin et al. (2001b) and Urban100 Huang et al. (2015) benchmarks. For image deraining, we utilize the Rain100L Yang et al. (2019) dataset, which contains 200 clean-rainy image pairs for training and 100 pairs for testing. For image classical SR, we choose DIV2K Agustsson & Timofte (2017) and Flickr2K Timofte et al. (2017) as the training set, and we evaluate on Set5 Bevilacqua et al. (2012), Set14 Zeyde et al. (2012), BSDS100 Martin et al. (2001a), and Urban100 Huang et al. (2015).

**Implementation Details and Evaluation.** All experiments were run on an NVIDIA Quadro RTX 4090 GPU using PyTorch. FraIR models with SVD value 350 were trained for 10k iterations with

Table 1: PSNR and SSIM results on Urban100 Huang et al. (2015) and BSD68 Martin et al. (2001b) for Gaussian denoising at noise levels $\sigma = 15$, 25, and 50. FraIR achieves the best performance across all settings. The best and the second-best results are highlighted in red and blue, respectively.

| Dataset | Urban100 | | | BSDS68 | | |
|---|---|---|---|---|---|---|
| Model | $\sigma = 15$ | $\sigma = 25$ | $\sigma = 50$ | $\sigma = 15$ | $\sigma = 25$ | $\sigma = 50$ |
| | PSNR / SSIM | PSNR / SSIM | PSNR / SSIM | PSNR / SSIM | PSNR / SSIM | PSNR / SSIM |
| DnCNN Zhang et al. (2017a) | 32.98 / 0.931 | 30.81 / 0.902 | 27.59 / 0.833 | 33.89 / 0.930 | 31.23 / 0.883 | 27.92 / 0.789 |
| IRCNN Zhang et al. (2017c) | 27.59 / 0.833 | 31.20 / 0.909 | 27.70 / 0.840 | 33.87 / 0.929 | 31.18 / 0.882 | 27.88 / 0.790 |
| FFDNet Zhang et al. (2018a) | 33.83 / 0.942 | 31.40 / 0.912 | 28.05 / 0.848 | 33.87 / 0.929 | 31.21 / 0.882 | 27.96 / 0.789 |
| BRDNet Tian et al. (2020) | 34.42 / 0.946 | 31.99 / 0.919 | 28.56 / 0.858 | 34.10 / 0.929 | 31.43 / 0.885 | 28.16 / 0.794 |
| AirNet Li et al. (2022a) | 34.40 / 0.949 | 32.10 / 0.924 | 28.88 / 0.871 | 34.14 / 0.936 | 31.48 / 0.893 | 28.23 / 0.806 |
| PromptIR Potlapalli et al. (2023) | 34.77 / 0.952 | 32.49 / 0.929 | 29.39 / 0.881 | 34.34 / 0.938 | 31.71 / 0.897 | 28.49 / 0.813 |
| AdaIR Cui et al. (2025) | 34.96 / 0.953 | 32.74 / 0.931 | 29.70 / 0.885 | 34.36 / 0.938 | 31.72 / 0.897 | 28.49 / 0.813 |
| Baseline Liang et al. (2021) | 35.13 / 0.953 | 32.90 / 0.932 | 29.82 / 0.885 | 34.42 / 0.931 | 31.78 / 0.878 | 28.56 / 0.754 |
| FraIR | 35.28 / 0.955 | 33.12 / 0.932 | 30.08 / 0.887 | 35.31 / 0.934 | 32.43 / 0.885 | 28.68 / 0.769 |

Adam Kingma (2014) and L1 loss, starting with a learning rate of $1e^{-3}$, reduced by 25% at 5k, 7.5k, and 9k iterations. In comparison, full model fine-tuning required 100k iterations to achieve comparable results, with the learning rate halved at 50k, 75k, and 90k intervals. Both FraIR and fine-tuning used the same settings, including the Adam optimizer Kingma (2014), a batch size of 4, and the L1 loss function. To evaluate fidelity and perceptual quality, we used PSNR and SSIM (Y channel) and perceptual metrics LPIPS Zhang et al. (2018b) and DISTS Ding et al. (2020).

**Backbones and Adaptation Setup.** To assess FraIR's adaptation capacity, we integrate it into two transformer-based backbones: SwinIR Liang et al. (2021) for denoising, deraining and EDT Li et al. (2023) for super-resolution and hybrid settings, representing efficient and hierarchical vision transformers. Both use official pre-trained weights, with FraIR added as a plug-in adapter. Training is conducted on a single NVIDIA RTX 4090 for 10,000 iterations using Adam, starting with a $1e^{-3}$ learning rate and decaying by 0.75 at 5000, 7500, and 9000 iterations.

**Baselines and Existing Methods.** We compare FraIR with several state-of-the-art efficient pre-trained model adaptation methods: (i) Pretrained (no adaptation), (ii) Full fine-tuning (Full-ft), (iii) SSF Lian et al. (2022) (learnable scale and shift on frozen features), (iv) VPT Jia et al. (2022b) (learnable prompts at transformer inputs), (v) Adapter Houlsby et al. (2019a) (bottlenecks after attention/MLP), (vi) LoRA Hu et al. (2022) (low-rank updates to query/value projections), (vii) AdaptFormer Chen et al. (2022a) (parallel MLP adaptation), (viii) FacT Jie & Deng (2023) (tensorized transformer with low-rank adaptation), and (ix) AdaptIR Guo et al. (2024) (MoE structure for spatial and channel-wise features). These comparisons highlight FraIR's superior efficiency and effectiveness. A conceptual overview of PETL is in Appendix A.1.

## 4.2 COMPARISON WITH STATE-OF-THE-ART METHODS

**Results on Denoising.** We evaluate FraIR on Gaussian denoising for noise levels $\sigma = 15$, 25, and 50 using Urban100 Huang et al. (2015) and BSD68 Martin et al. (2001b). All denoising methods—including all-in-one models—are compared under their single-task training setting, rather than their all-in-one versions. As shown in Table 1, FraIR consistently outperforms prior methods in terms of PSNR, achieving the best results across all noise levels. On Urban100, FraIR achieves 35.28 dB at $\sigma = 15$, a 0.92% improvement over AdaIR, the strongest baseline. At $\sigma = 25$ and 50, FraIR outperforms AdaIR by 1.16% and 1.28%, respectively. SSIM values are also highest at $\sigma = 15$ and 25, improving over AdaIR by 0.21% and 0.11%, respectively. On BSD68, FraIR similarly leads in PSNR, improving over AdaIR by 2.77%, 2.23%, and 0.67% at increasing noise levels. Furthermore, the bottom part of Figure 3 illustrates its ability to recover fine structures such as text and edges, where competing methods fail to fully remove noise artifacts. Overall, FraIR delivers robust denoising performance, surpassing both traditional and recent adaptive methods, validating the strength of frequency-domain adaptation for image restoration.

**Results on Deraining.** We evaluate our method on the Rain100L dataset to assess its effectiveness in a dedicated deraining task. As presented in Table 2, FraIR achieves a PSNR of 38.38dB, outperforming the PEFT-based AdaptIR by a significant margin of 0.57dB. Importantly, FraIR also consistently surpasses all-in-one restoration models such as PromptIR and DaAIR—even though these models are trained solely for the deraining task in this setting, rather than in a multi-task or general-purpose

Table 2: Deraining results in the single-task setting on Rain100L Yang et al. (2019). Compared to the baseline SwinIR Liang et al. (2021) and the state-of-the-art AdaIR Cui et al. (2025), the proposed approach yields 1.5 dB and 0.3 dB PSNR improvement, respectively.

| Method | UMRL | MSPFN | LPNet | AirNet | Restormer | PromptIR | DaAIR | AdaptIR | AdaIR | Baseline | FraIR |
|--------|------|-------|-------|--------|-----------|----------|-------|---------|-------|----------|-------|
| PSNR | 32.39 | 33.50 | 33.61 | 34.90 | 36.74 | 37.04 | 37.78 | 37.81 | 38.34 | 36.84 | **38.38** |
| SSIM | 0.921 | 0.948 | 0.958 | 0.977 | 0.978 | 0.979 | 0.982 | 0.981 | **0.983** | 0.974 | 0.981 |

Table 3: Quantitative results on five benchmark datasets for single image super-resolution at ×2, ×3, and ×4 scales. Our method, FraIR, achieves the best performance across most settings while maintaining the lowest number of trainable parameters. Compared to full fine-tuning and recent PETL methods under the same training setting with EDT backbone Li et al. (2023), FraIR demonstrates superior accuracy and efficiency.

| Scale | Model | Params. | Set5 | Set14 | BSD100 | Urban100 | Manga109 |
|-------|-------|---------|------|-------|--------|----------|----------|
| x2 | Baseline | - | 38.28 | 34.46 | 32.48 | 33.68 | 39.57 |
| | Fine-Tuning | 11.8M | 38.39 | 34.56 | 32.54 | 33.98 | 39.90 |
| | VPT Jia et al. (2022b) | 884K | 38.35 | 34.47 | 32.48 | **34.74** | 39.77 |
| | Adapter Houlsby et al. (2019a) | 691K | 38.36 | 34.50 | 32.48 | 33.73 | 39.80 |
| | LoRA Hu et al. (2022) | 995K | 38.38 | 34.52 | 32.49 | 33.76 | 39.80 |
| | AdaptFormer Chen et al. (2022a) | 677K | 38.37 | 34.48 | 32.48 | 33.72 | 39.79 |
| | SSF Lian et al. (2022) | 373K | 37.47 | 33.06 | 31.82 | 30.85 | 37.46 |
| | Fact Jie & Deng (2023) | 537K | 38.38 | 34.52 | 32.49 | 33.76 | 39.83 |
| | AdaptIR Guo et al. (2024) | 370K | 38.38 | 34.51 | 32.49 | 33.77 | 39.82 |
| | FraIR (Ours) | 347k | **38.46** | **34.64** | **32.59** | 34.01 | **40.28** |
| x3 | Baseline | - | 34.66 | 30.80 | 29.32 | 29.39 | 34.38 |
| | Fine-Tuning | 12M | 34.84 | **30.97** | 29.41 | 29.67 | 34.91 |
| | VPT Jia et al. (2022b) | 884K | 34.78 | 30.86 | 29.37 | 29.44 | 34.76 |
| | Adapter Houlsby et al. (2019a) | 691K | 34.78 | 30.88 | 29.37 | 29.46 | 34.83 |
| | LoRA Hu et al. (2022) | 995K | 34.79 | 30.88 | 29.38 | 29.49 | 34.83 |
| | AdaptFormer Chen et al. (2022a) | 677K | 34.78 | 30.88 | 29.37 | 29.45 | 34.81 |
| | SSF Lian et al. (2022) | 373K | 33.74 | 29.90 | 28.80 | 27.25 | 32.33 |
| | Fact Jie & Deng (2023) | 537K | 34.79 | 30.89 | 29.38 | 29.50 | 34.86 |
| | AdaptIR Guo et al. (2024) | 370K | 34.80 | 30.89 | 29.38 | 29.48 | 34.86 |
| | FraIR (Ours) | 347k | **34.88** | **30.90** | **29.42** | **29.78** | **35.40** |
| x4 | Baseline | - | 32.58 | 28.97 | 27.79 | 27.18 | 31.41 |
| | Fine-Tuning | 11.9M | 32.66 | 29.03 | 27.82 | 27.31 | 31.64 |
| | VPT Jia et al. (2022b) | 884K | 32.70 | 29.02 | 27.82 | 27.20 | 31.65 |
| | Adapter Houlsby et al. (2019a) | 691K | 32.70 | 29.03 | 27.82 | 27.21 | 31.68 |
| | LoRA Hu et al. (2022) | 995K | 32.70 | 29.03 | 27.82 | 27.22 | 31.68 |
| | AdaptFormer Chen et al. (2022a) | 677K | 32.70 | 29.03 | 27.82 | 27.21 | 31.68 |
| | SSF Lian et al. (2022) | 373K | 31.57 | 28.20 | 27.30 | 25.38 | 29.36 |
| | Fact Jie & Deng (2023) | 537K | 32.71 | 29.03 | 27.83 | 27.23 | 31.70 |
| | AdaptIR Guo et al. (2024) | 370K | 32.71 | **29.04** | 27.82 | 27.22 | 31.70 |
| | FraIR (Ours) | 347k | **32.80** | 29.00 | **27.94** | **27.49** | **32.32** |

manner. Furthermore, the upper part of Figure 3 illustrates the effectiveness of FraIR in recovering fine textures, such as those around the eyes and hair, where competing methods fail to fully remove rain artifacts. This substantial performance gain highlights FraIR's ability to effectively specialize and adapt, even when compared to larger or purpose-tuned architectures. Unlike heavy all-in-one designs, FraIR achieves superior restoration with a lightweight, modular architecture.

**Results on Super-Resolution.** Table 3 presents the super-resolution performance across various upscaling factors. Our method achieves superior results compared to the current state-of-the-art, FacT Jie & Deng (2023) across most datasets and scales, while maintaining the lowest parameter count. Notably, under the ×4 scale, it surpasses even full fine-tuning in performance, despite utilizing only 0.3% of the trainable parameters. Additionally, we observe that the recently introduced SSF Lian et al. (2022) performs poorly across all benchmarks. This can be attributed to its reliance on channel-wise transformations, which neglect spatial information that is critical in low-level restoration tasks. Moreover, compared to AdaptIR Guo et al. (2024), which already offers a competitive balance between performance and efficiency, our method provides further improvements in both accuracy and parameter efficiency. These results highlight the limitations of directly applying techniques developed for high-level vision tasks to restoration scenarios, where spatial fidelity plays a more significant role.

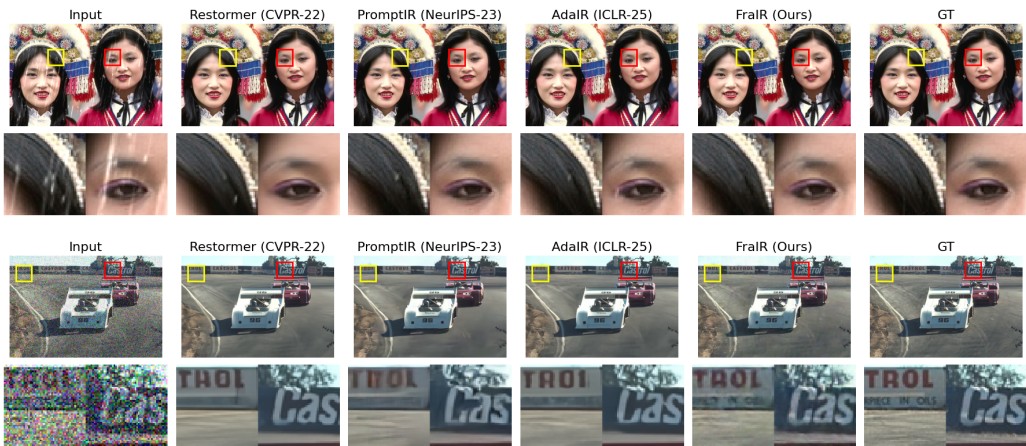

Figure 3: For qualitative results, we compare FraIR to Restormer Zamir et al. (2022), PromptIR Potlapalli et al. (2023) and AdaIR Cui et al. (2025) for different degradations. FraIR effectively removes rain streaks and noise injections while preserving image sharpness, achieving high-quality restoration.

Table 4: Performance comparison under two hybrid degradations—(i) ×4 downsampling with Gaussian noise ($\sigma = 30$) and (ii) ×4 downsampling with JPEG compression (quality factor = 30)—across five benchmarks. Most PETL methods exhibit notable degradation under these heterogeneous conditions, whereas FraIR maintains strong generalization and restoration quality.

| Hybrid Degradation | Model | Trainable Params. | Set5 PSNR | Set5 SSIM | Set14 PSNR | Set14 SSIM | BSD100 PSNR | BSD100 SSIM | Urban100 PSNR | Urban100 SSIM | Manga109 PSNR | Manga109 SSIM |
|---|---|---|---|---|---|---|---|---|---|---|---|---|
| | Baseline | | 19.74 | 0.3569 | 19.27 | 0.3114 | 19.09 | 0.2783 | 18.54 | 0.3254 | 19.75 | 0.3832 |
| | SSF Lian et al. (2022) | 373K | 25.41 | 0.6720 | 24.02 | 0.5761 | 24.06 | 0.5411 | 21.89 | 0.5514 | 23.33 | 0.6736 |
| | VPT Jia et al. (2022b) | 884K | 24.11 | 0.5570 | 22.97 | 0.4722 | 22.91 | 0.4336 | 21.20 | 0.4527 | 22.61 | 0.5570 |
| | Adapter Houlsby et al. (2019a) | 691K | 25.60 | 0.6862 | 24.16 | 0.5856 | 24.17 | 0.5498 | 22.05 | 0.5640 | 23.61 | 0.6904 |
| LR4 & Noise30 | LoRA Hu et al. (2022) | 995K | 25.19 | 0.6371 | 23.82 | 0.5405 | 23.82 | 0.5026 | 21.81 | 0.5193 | 23.30 | 0.6396 |
| | AdaptFormer Chen et al. (2022a) | 677K | 26.10 | 0.7138 | 24.58 | 0.6095 | 24.44 | 0.5686 | 22.52 | 0.5976 | 24.38 | 0.7296 |
| | FacT Jie & Deng (2023) | 537K | 25.70 | 0.6963 | 24.24 | 0.5944 | 24.25 | 0.5586 | 21.10 | 0.5727 | 23.63 | 0.6993 |
| | FourierFT Gao et al. (2024) | 582K | 26.40 | 0.7420 | 24.80 | 0.6320 | 24.60 | 0.6200 | 22.75 | 0.5900 | 24.90 | 0.7600 |
| | AdaptIR Guo et al. (2024) | 697K | 26.48 | 0.7441 | 24.88 | 0.6345 | 24.67 | 0.6279 | 22.88 | 0.5932 | 24.96 | 0.7625 |
| | FraIR | 347k | 26.81 | 0.7575 | 25.05 | 0.6416 | 24.76 | 0.5920 | 23.37 | 0.6523 | 25.76 | 0.7881 |
| | Baseline | | 25.08 | 0.6638 | 23.95 | 0.5847 | 21.51 | 0.5569 | 24.08 | 0.5580 | 22.60 | 0.6612 |
| | SSF Lian et al. (2022) | 373K | 26.91 | 0.7664 | 25.33 | 0.6502 | 23.21 | 0.6519 | 24.98 | 0.6027 | 24.98 | 0.7801 |
| | VPT Jia et al. (2022b) | 884K | 26.39 | 0.7367 | 24.84 | 0.6306 | 22.48 | 0.6101 | 24.75 | 0.5902 | 23.98 | 0.7365 |
| | Adapter Houlsby et al. (2019a) | 691K | 27.00 | 0.7698 | 25.36 | 0.6518 | 23.30 | 0.6566 | 25.01 | 0.6039 | 25.13 | 0.7848 |
| LR4 & JPEG30 | LoRA Hu et al. (2022) | 995K | 27.01 | 0.7694 | 25.36 | 0.6513 | 23.26 | 0.6551 | 25.00 | 0.6038 | 25.09 | 0.7837 |
| | AdaptFormer Chen et al. (2022a) | 677K | 27.03 | 0.7715 | 25.40 | 0.6533 | 23.32 | 0.6581 | 25.02 | 0.6048 | 25.19 | 0.7873 |
| | FacT Jie & Deng (2023) | 537K | 27.01 | 0.7703 | 25.37 | 0.6521 | 23.30 | 0.6569 | 25.00 | 0.6041 | 25.14 | 0.7855 |
| | FourierFT Gao et al. (2024) | 582K | 27.05 | 0.7700 | 25.32 | 0.6510 | 23.30 | 0.6580 | 24.95 | 0.6000 | 25.22 | 0.7880 |
| | AdaptIR Guo et al. (2024) | 697K | 27.13 | 0.7739 | 25.44 | 0.6545 | 23.41 | 0.6620 | 25.04 | 0.6057 | 25.29 | 0.7903 |
| | FraIR | 348k | 27.22 | 0.7786 | 25.68 | 0.6602 | 23.69 | 0.6698 | 25.18 | 0.6725 | 25.37 | 0.7922 |

## 4.3 COMPARISON ON HYBRID DEGRADATION TASKS

Evaluating PETL methods separately on each degradation type can be costly and may fail to reveal their ability to generalize beyond narrowly defined corruption models. To provide a more challenging and representative benchmark, we evaluate all methods under hybrid degradations that combine heterogeneous corruption sources. In particular, Table 4 reports results for two second-order hybrid settings: (i) ×4 downsampling with additive Gaussian noise ($\sigma$=30), denoted as LR4 & Noise30 and (ii) ×4 downsampling with JPEG compression (quality factor = 30), denoted as LR4 & JPEG30. Both settings require PETL methods to model mixed degradations with distinct frequency characteristics, making them a strong test of adaptation generalization. As shown in Table 4, FraIR consistently outperforms all PETL baselines across five benchmarks. Under LR4 & Noise30, FraIR achieves 23.37 dB on Urban100 and 25.76 dB on Manga109, outperforming AdaptIR by 0.49 dB and 0.80 dB, respectively. Under LR4 & JPEG30, FraIR again delivers the highest PSNR and SSIM across all datasets, surpassing AdaptIR by up to 0.25 dB on Set14 and 0.11 dB on Urban100, while using less than half the number of tunable parameters. These results confirm that FraIR effectively captures heterogeneous degradations and generalizes reliably under complex real-world corruption mixtures. Additional hybrid-degradation results are provided in the supplementary material.

Table 5: Denoising performance of FraIR with EDT and SwinIR backbones under different Gaussian noise levels ($\sigma = 15, 25, 50$).

| Backbone | $\sigma = 15$ | | | $\sigma = 25$ | | | $\sigma = 50$ | | |
|---|---|---|---|---|---|---|---|---|---|
| | PSNR | SSIM | LPIPS | PSNR | SSIM | LPIPS | PSNR | SSIM | LPIPS |
| EDT Li et al. (2023) | 35.31 | 0.934 | 0.0996 | 32.43 | 0.885 | 0.1256 | 28.68 | 0.769 | 0.2216 |
| SwinIR Liang et al. (2021) | 36.12 | 0.947 | 0.0576 | 33.37 | 0.909 | 0.0975 | 29.95 | 0.831 | 0.1925 |

## 4.4 ABLATION STUDIES

**Generalization Across Restoration Backbones.** To demonstrate the modularity and robustness of FraIR, we integrate it into two distinct pre-trained Transformer-based backbones: EDT Li et al. (2023) and SwinIR Liang et al. (2021). Without modifying backbone weights, FraIR consistently enhances performance across noise levels while adapting only 1–3% of total parameters. As shown in Table 5, FraIR-equipped SwinIR outperforms EDT across all metrics, particularly under severe noise ($\sigma = 50$), highlighting its capacity to complement stronger backbones. These results confirm that FraIR generalizes well across architectures, reinforcing its role as a lightweight, frequency-aware adapter for scalable image restoration. More comparative results for backbones are provided in Appendix A.5.

**Runtime and FFT Overhead.** To quantify efficiency, we measure training and inference costs against other PETL baselines on a single RTX 4090 (batch size 1, FP32). Results in Table 6 show that FraIR introduces only a modest +5.4% FLOPs during training (6.0 ms/image vs. 5.7 ms for the frozen backbone), while maintaining the fastest training time among PETLs. Crucially, after weight fusion the FFT overhead disappears entirely, yielding identical inference latency to the frozen backbone (5.7 ms). Competing PETLs retain 5–8% extra cost at inference. This demonstrates that FraIR achieves parameter-efficient adaptation without compromising deployment speed, offering a favorable balance between spectral modeling capacity and real-time feasibility.

Table 6: Runtime and FLOPs comparison at $512 \times 512$ and all benchmarks are measured on a single RTX 4090 (batch size 1, FP32).

| Method | Extra Params | Train (ms) | Inference (ms) | +FLOPs |
|---|---|---|---|---|
| Frozen SwinIR | 0% | 5.7 | 5.7 | 0% |
| ARC Dong et al. (2024) | 0.58% | 6.3 | 6.1 | +6.6% |
| LoRA ($r = 8$) | 0.92% | 6.2 | 6.0 | +6.5% |
| FourA Borse et al. (2024) | 0.75% | 6.4 | 6.2 | +6.8% |
| FraIR (ours) | 0.45% | 6.0 | 5.7 | +5.4% |

**Controlling Expressiveness via SVD Rank.** FraIR employs low-rank spectral projections to balance adaptation capacity and efficiency. To assess this, we vary the SVD rank $r$ on Fourier-transformed features and observe its impact in Table 7. Performance improves consistently up to $r = 500$—e.g., PSNR rises from 34.59 to 35.40, while LPIPS drops from 0.0635 to 0.0515 thus indicating richer frequency-aware adaptation. Gains saturate or slightly decline at $r = 1000$, suggesting overfitting or redundancy. These results demonstrate the tunable expressiveness of our Fourier adapter, enabling strong performance even at $r = 350$ with under 800k trainable parameters, well below full fine-tuning thus making it ideal for efficient restoration.

## 5 DISCUSSION

**How effective is the Fourier Adapter?** To evaluate the impact of adapting in the frequency domain, we compare FraIR with four structurally related baselines: (i) ARC Dong et al. (2024), which applies a shared low-rank spatial projection $U\Lambda U^\top$; (ii) GARC, which adds a gating mechanism to ARC, serves as a spatial-domain analogue of our design, (iii) a Wavelet Adapter operating in the 1D Haar wavelet domain; and (iv) FouRA, which performs fixed Fourier-domain modulation without learnable low-rank recomposition. All methods use comparable parameter budgets and ranks, but only FraIR performs continuous 1D Fourier adaptation with learnable spectral decomposition, low-rank recomposition, and dynamic reweighting. As shown in Figure 4, FraIR consistently outperforms all baselines on ×4 SR realSR dataset Cai et al. (2019), achieving higher fidelity and better perceptual

quality. Unlike spatial, wavelet, or fixed Fourier adapters, FraIR captures global spectral interactions with task-aware modulation, demonstrating the advantages of frequency-domain PETL. Additional results are provided in the supplementary material.

**Limitations.** While FraIR shows strong performance in real-world image restoration, it has several limitations. First, 2D Fourier Transform is constrained by GPU memory, limiting full-resolution spectral learning; thus, the current design uses 1D Fourier operations, which may reduce spatial-frequency interaction modeling. Second, although FRA is parameter-efficient and generalizes well, its frequency-based design may underperform on degradations lacking structured spectral patterns. Lastly, adapting FRA to compact or mobile architectures without sacrificing quality remains challenging.

Table 7: Ablation on SVD rank in the Fourier Adapter (Rain100L).

| Model | PSNR | SSIM | LPIPS | Params |
|---|---|---|---|---|
| svd50 | 33.37 | 0.9448 | 0.0948 | 125k |
| svd100 | 34.59 | 0.9558 | 0.0635 | 236k |
| svd150 | 34.42 | 0.9560 | 0.0617 | 347k |
| svd350 | 34.86 | 0.9593 | 0.0548 | 794k |
| svd500 | 35.40 | 0.9626 | 0.0515 | 1.1M |
| svd1000 | 35.10 | 0.9617 | 0.0523 | 2.3M |

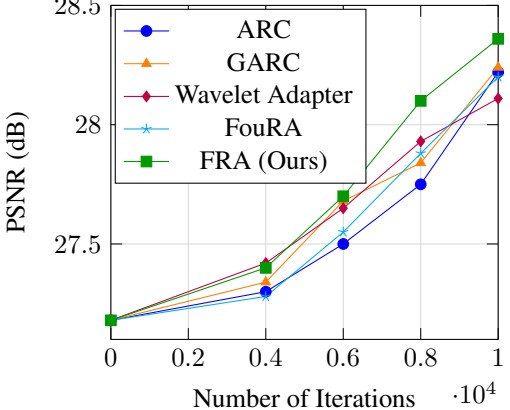

Figure 4: PSNR vs. training iterations for ARC, GARC, Wavelet Adapter, FouRA, and FRA. on RealSR Cai et al. (2019) dataset with the same baseline.

## 6 CONCLUSION

We introduce **FraIR**, a novel frequency-domain adapter for parameter-efficient image restoration. By decomposing token representations into spectral components and applying low-rank, learnable modulation in Fourier space, FraIR bridges the gap between efficiency and expressiveness. It enables task-adaptive restoration behavior using only a small set of trainable parameters, while preserving the backbone model's structure and knowledge. Unlike conventional PETL approaches that operate in the spatial domain, or Fourier-based methods focused on generative modeling or static compression, FraIR delivers dynamic, reweightable adaptation in the spectral domain. Our method integrates seamlessly into Transformer layers, supports reparameterization for inference-time efficiency, and generalizes well across both seen and unseen degradations. Extensive results demonstrate that FraIR consistently outperforms state-of-the-art PETL baselines and achieves comparable or superior performance to full fine-tuning at a fraction of the computational cost. This makes it a practical and principled solution for real-world image restoration applications.

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
