## A APPENDIX - FRAIR: FOURIER RECOMPOSITION ADAPTER FOR IMAGE RESTORATION

### A.1 OVERVIEW OF PARAMETER EFFICIENT TRANSFER LEARNING METHODS

We compare our method with a range of strong parameter-efficient tuning (PETL) baselines, each employing distinct adaptation mechanisms within Transformer architectures.

VPT Jia et al. (2022b), illustrated in Fig. 5 (a), introduces learnable prompt tokens prepended to the input of each Transformer layer. Among its two variants—VPT-Shallow, which adds prompts only at the input layer, and VPT-Deep, which inserts prompts into every layer—we adopt VPT-Deep as it yields stronger performance.

Adapter Houlsby et al. (2019a), shown in Fig. 5 (b), adopts a bottleneck structure with a GELU activation in between. Following its standard configuration, we insert adapters after both the Multi-Head Self-Attention (MHSA) and the MLP submodules in each Transformer block.

LoRA Hu et al. (2022), depicted in Fig. 5 (c), approximates task-specific updates by injecting low-rank trainable matrices into the query and value projections of the attention mechanism. It performs updates via matrix multiplications of two low-rank factors.

AdaptFormer Chen et al. (2022a), as visualized in Fig. 5 (d), structurally resembles Adapter but differs in placement and form. It is inserted in parallel to the MLP module and placed before the second LayerNorm layer, enabling more flexible tuning.

SSF (Scale and Shift Fine-tuning) Lian et al. (2022), shown in Fig. 5 (e), modulates frozen features using learnable scale and shift parameters. As in the original setup, we place SSF modules after the QKV projections, LayerNorm, and MLP layers.

FacT Jie & Deng (2023), illustrated in Fig. 5 (f), tensorizes Transformer weights using Tucker decomposition and applies low-rank updates. Unlike LoRA, FacT shares the up/down projection weights across layers while keeping low-rank projections layer-specific. Among its two variants, we use FacT-TT for its superior performance, inserting it into both attention and MLP layers.

AdaptIR Guo et al. (2024) employs a Mixture-of-Experts (MoE) adapter structure designed to capture heterogeneous information across local spatial, global spatial, and channel-wise dimensions. Each branch learns a distinct basis representation, and a learned combination mechanism adaptively merges these branches. AdaptIR modules are applied in both attention and MLP pathways and are known for their strong generalization across complex degradation patterns.

### A.2 ARCHITECTURE OVERVIEW

FraIR is a plug-and-play adapter module designed for seamless integration into Transformer-based image restoration networks. As illustrated in Figure 6, FraIR layers are inserted at two key points within each Transformer block: one before the Multi-Head Self-Attention (MHSA) module and another before the Feed-Forward Network (MLP) module. These two instances of FraIR operate independently, each with its own learnable parameters. During training, only the parameters of the FraIR modules are updated, while the rest of the backbone network remains frozen. This parameter-efficient tuning strategy enables rapid adaptation to new tasks or domains without requiring full fine-tuning of the underlying model.

### A.3 MORE RESULTS ON HYBRID DEGRADATIONS

Beyond super-resolution, we further demonstrate FraIR's versatility on restoration tasks with compound degradations. Table 8 shows results on Rain100L with added Gaussian noise ($\sigma = 50$). FraIR outperforms specialized restoration models such as AirNet, PromptIR, and AdaIR, achieving a PSNR of 27.82 dB and SSIM of 0.813, marking the best performance among all methods. These results highlight FraIR's ability to effectively generalize across restoration tasks with spatially and spectrally diverse degradation types.

(a) VPT

(b) Adapter

(c) LoRA

(d) AdaptrFormer

(e) SSF

(f) FacT

(g) AdaptIR

(h) FraIR

Figure 5: Architectural comparison of PETL methods, including VPT, Adapter, LoRA, AdaptFormer, SSF, FacT, and AdaptIR.

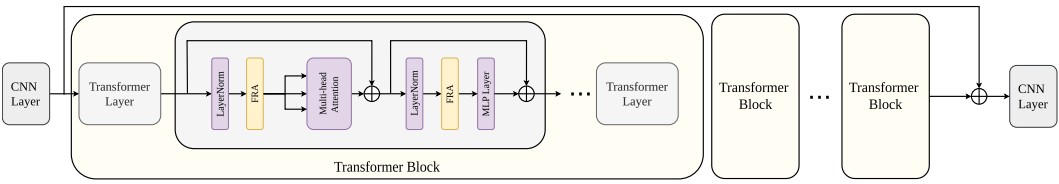

Figure 6: Integration of FraIR into Transformer blocks. FraIR modules are inserted before both the Multi-Head Self-Attention and MLP layers, enabling task-specific adaptation while keeping the rest of the backbone frozen.

Table 8: Results on mixed degradations, Rain100L with the Gaussian noise $\sigma= 50$.

| Method | AirNet | PromptIR | AdaIR | FraIR (Ours) |
|--------|--------|----------|-------|--------------|
| PSNR   | 27.25  | 27.34    | 27.51 | **27.82**    |
| SSIM   | 0.790  | 0.791    | 0.799 | **0.813**    |

Table 9: *Quantitative comparison of our proposed FraIR with the state-of-the-art real-world GAN and diffusion methods*. Our approach achieves high PSNR and SSIM scores while also delivering perceptual scores comparable to GAN and diffusion-based methods trained with perceptual losses.

| | Method | Trainable Parameters | RealSR | | | | DRealSR | | | |
|---|---|---|---|---|---|---|---|---|---|---|
| | | | PSNR | SSIM | LPIPS | DISTS | PSNR | SSIM | LPIPS | DISTS |
| GAN | RealESRGAN Wang et al. (2021) | 17M | 25.69 | 0.7616 | 0.2727 | 0.2063 | 28.64 | 0.8053 | 0.2847 | 0.2089 |
| | LDL Liang et al. (2022) | 12M | 24.96 | 0.7634 | 0.2519 | 0.1981 | 27.43 | 0.8078 | 0.2655 | 0.2055 |
| | SAFMN-L Sun et al. (2023) | 5.6M | 24.23 | 0.7217 | 0.2905 | 0.2176 | 27.15 | 0.7671 | 0.3148 | 0.2219 |
| Diffusion | StableSR Yang et al. (2024) | 150M | 24.70 | 0.7085 | 0.3018 | 0.2135 | 28.13 | 0.7542 | 0.3315 | 0.2263 |
| | ResShift Yue et al. (2023) | 119M | 26.31 | 0.7421 | 0.3460 | 0.2498 | 28.46 | 0.7673 | 0.4006 | 0.2656 |
| | SeeSR Wu et al. (2024b) | 750M | 25.18 | 0.7216 | 0.3009 | 0.2223 | 28.17 | 0.7691 | 0.3189 | 0.2315 |
| | PASD Yang et al. (2024) | 625M | 25.21 | 0.6798 | 0.3380 | 0.2260 | 27.36 | 0.7073 | 0.3760 | 0.2531 |
| | DiffBIR Lin et al. (2025) | 380M | 24.75 | 0.6567 | 0.3636 | 0.2312 | 26.71 | 0.6571 | 0.4557 | 0.2748 |
| | SinSR Wang et al. (2024b) | 119M | 26.28 | 0.7347 | 0.3188 | 0.2353 | 28.36 | 0.7515 | 0.3665 | 0.2485 |
| | OSEDiff Wu et al. (2024a) | 8.5M | 25.15 | 0.7341 | 0.2921 | 0.2128 | 27.92 | 0.7835 | 0.2968 | 0.2165 |
| PETL | DiffFit Xie et al. (2023) | 13k | 28.27 | 0.7949 | 0.2949 | 0.2199 | 30.78 | 0.8384 | 0.3617 | 0.2541 |
| | ARC Dong et al. (2024) | 906k | 28.09 | 0.7924 | 0.2760 | 0.2144 | 30. 77 | 0.8392 | 0.3485 | 0.2516 |
| | AdaptSR Korkmaz et al. (2025) | 886k | **28.70** | **0.8079** | **0.2591** | **0.2109** | 30.74 | 0.8422 | 0.3381 | **0.2524** |
| | FraIR | 347k | 28.66 | 0.7933 | 0.2690 | 0.2125 | **30.82** | **0.8499** | **0.3321** | 0.2941 |

## A.4 EXTENSION TO REALSR

To evaluate FraIR on real-world super-resolution, we follow the standard RealSR adaptation protocol using the DIV2K Agustsson & Timofte (2017) and RealSR Cai et al. (2019) and DRealSR Wei et al. (2020) training sets. Training is performed on randomly cropped $256 \times 256$ patches from high-resolution images. For evaluation, we adopt the validation setup from StableSR Wang et al. (2024a), which includes 100 $128 \times 128$ patches from RealSR and 93 from DRealSR, each paired with corresponding $512 \times 512$ high-resolution targets.

Quantitative results are reported in Table 9. Compared to state-of-the-art GAN and diffusion-based models—many of which are trained with perceptual losses—FraIR achieves significantly stronger performance in terms of distortion-oriented metrics such as PSNR and SSIM, while maintaining competitive perceptual quality. On RealSR, FraIR achieves a PSNR of 28.66 dB with only 347K trainable parameters, outperforming nearly all diffusion-based methods including SeeSR (25.18 dB) and PASD (25.21 dB), and even surpassing the 150M-parameter StableSR model (24.70 dB). On DRealSR, FraIR achieves the highest PSNR (30.82 dB) and SSIM (0.8499), highlighting its robustness and generalization to diverse real-world degradations.

Among PETL baselines, FraIR matches or exceeds the performance of AdaptSR and ARC while requiring significantly fewer parameters. Notably, it achieves comparable fidelity to AdaptSR (28.70 dB vs. 28.66 dB on RealSR), while improving SSIM on DRealSR by a noticeable margin (0.8499 vs. 0.8422), demonstrating its effectiveness in scaling to realistic degradation patterns with minimal overhead.

## A.5 FURTHER COMPARISON FOR BACKBONES

To assess the compatibility of FraIR with different backbone architectures, we conduct additional experiments on the Rain100L dataset using two representative restoration backbones: EDT and SwinIR. As shown in Table 10, both configurations achieve strong performance, validating the generality of our adapter design. Specifically, the FraIR-equipped EDT model yields 38.38 dB PSNR and 0.981 SSIM, while SwinIR achieves slightly higher fidelity with 38.45 dB PSNR and 0.982 SSIM. This indicates that FraIR seamlessly integrates into different architectures and effectively enhances performance across varying design choices. The consistent improvement across both backbones further underscores FraIR's adaptability and robustness in modeling complex degradations like rain streaks.

Table 10: Deraining performance comparison between EDT and SwinIR backbones on Rain100L. SwinIR achieves slightly better results than EDT.

| Backbone | PSNR ↑ | SSIM ↑ |
|----------|--------|--------|
| EDT | 38.38 | 0.981 |
| SwinIR | **38.45** | **0.982** |

Table 11: Comparison of adapter variants on the Rain100L dataset for single-task image deraining. FraIR achieves the best performance in both PSNR and SSIM.

| Method | PSNR ↑ | SSIM ↑ |
|--------|--------|--------|
| ARC Adapter | 37.64 | 0.977 |
| Gated-ARC Adapter | 37.86 | 0.978 |
| Wavelet Adapter | 37.87 | 0.978 |
| Foura | 38.02 | 0.977 |
| FraIR (Ours) | **38.38** | **0.981** |

## A.6 FRAIR ON CNN BACKBONES

Although FraIR was primarily evaluated on Transformer backbones (e.g., SwinIR, EDT), it can also be applied to CNN architectures. Specifically, each 2D convolution weight $W$ is replaced by its frequency-domain FraIR update, with stride and padding preserved. This requires no architectural changes and introduces only a small overhead in trainable parameters.

We demonstrate this on two representative CNN or hybrid models: **Restormer** (CNN–Transformer hybrid) for deraining, and **NAFNet** (lightweight CNN) for denoising. Results are summarized in Table 12.

Table 12: FraIR applied to Transformer and CNN backbones. FraIR consistently outperforms LoRA and FouRA while requiring fewer extra parameters.

| Backbone & Task | Adapter | PSNR ↑ | SSIM ↑ | Extra Params |
|-----------------|---------|--------|--------|--------------|
| Restormer (Rain1400, derain) | LoRA ($r$=8) | 35.42 | 0.960 | 0.87% |
| | FouRA | 35.63 | 0.962 | 0.71% |
| | FraIR (ours) | **35.85** | **0.965** | **0.49%** |
| NAFNet (SIDD, denoise) | LoRA ($r$=8) | 39.03 | 0.913 | 0.87% |
| | FouRA | 39.18 | 0.914 | 0.71% |
| | FraIR (ours) | **39.35** | **0.916** | **0.49%** |

These results confirm that FraIR extends beyond Transformers and provides consistent gains on CNN-based architectures with even lower parameter overhead than prior PETLs.

## A.7 OBSERVATION ON LEARNED FREQUENCY GATES

We average FraIR's learned frequency weights over 200 validation images for denoising, deblurring, and deraining.

- Denoising boosts mid-band (40–70%), suppresses high frequencies.
- Deblurring amplifies top 15% to restore high-band detail.
- Deraining shows dual peaks (low-mid, high) capturing streak and veil.

Cosine similarity between task gates: blur vs. noise = 0.17, blur vs. rain = 0.23, noise vs. rain = 0.42.

## A.8 1D VS 2D FFT ABLATION

We compare FraIR with 1D and 2D FFTs on $64 \times 64$ patches (SwinIR, Rain100H).

Although 2D FFT provides +0.12 dB, its memory cost exceeds 24 GB at $256^2$. Therefore we adopt 1D FFT as default.

## A.9 ADAPTER COMPARISON

To assess the effectiveness of our proposed Fourier Adapter, we conduct ablation studies comparing it against several alternative adapter designs under identical training settings. These include: (i) the

Table 13: 1D vs 2D FFT.

| FFT mode | PSNR | VRAM | Iter-time (ms) | Overhead |
|----------|------|------|----------------|----------|
| 1D | 34.57 | 0.6 GB | 10.1 | – |
| 2D | 34.69 | 1.3 GB | 17.4 | $\times$2.17 VRAM, $\times$1.72 time |

ARC adapter Dong et al. (2024), (ii) its gated extension (GARC), and (iii) a Wavelet-based adapter leveraging the Haar Discrete Wavelet Transform (DWT). All variants aim to introduce lightweight task-specific modulation into the frozen backbone, and their architectural differences are isolated for fair comparison.

**ARC Adapter.** The ARC adapter inserts a low-rank modulation into the residual stream by applying a shared projection in both directions. Given an input hidden state $\mathbf{X} \in \mathbb{R}^{B \times L \times D}$, the ARC adapter is defined as:

$$\text{ARC}(\mathbf{X}) = \mathbf{X} + \left( (\mathbf{X} \cdot (\mathbf{U} \circ \mathbf{\Lambda})) \cdot \mathbf{U}^\top + \mathbf{b} \right)$$

where $\mathbf{U} \in \mathbb{R}^{D \times r}$ is the shared projection matrix, $\mathbf{\Lambda} \in \mathbb{R}^r$ is a learnable scaling vector, $\circ$ denotes element-wise multiplication across columns, $\mathbf{b} \in \mathbb{R}^D$ is a learnable bias.

**Gated ARC Adapter (GARC).** The GARC adapter extends ARC by introducing a learnable gate $\mathbf{g} \in \mathbb{R}^D$ to control the strength of the residual modulation dynamically. It is formulated as:

$$\text{GARC}(\mathbf{X}) = \mathbf{X} + \mathbf{g} \circ \left( (\mathbf{X} \cdot (\mathbf{U} \circ \mathbf{\Lambda})) \cdot \mathbf{U}^\top + \mathbf{b} \right)$$

This design allows the model to learn per-channel modulation strengths, potentially improving flexibility over the vanilla ARC design.

**Wavelet Adapter.** In this variant, the adapter operates in the wavelet domain. The input feature $\mathbf{X}$ is first transformed via a 1D Discrete Wavelet Transform $\mathcal{W}$, and then the GARC adapter is applied to the approximation coefficients. The result is mapped back to the spatial domain using the inverse wavelet transform $\mathcal{W}^{-1}$. The formulation is:

$$\tilde{\mathbf{X}} = \mathcal{W}(\mathbf{X})$$
$$\mathbf{Z} = (\tilde{\mathbf{X}} \cdot (\mathbf{U} \circ \mathbf{\Lambda})) \cdot \mathbf{U}^\top + \mathbf{b}$$
$$\hat{\mathbf{X}} = \mathcal{W}^{-1}(\mathbf{g} \circ \mathbf{Z})$$

$$\text{WaveletAdapter}(\mathbf{X}) = \mathbf{X} + \hat{\mathbf{X}}$$

Here, $\mathcal{W}$ and $\mathcal{W}^{-1}$ represent the 1D Haar wavelet decomposition and reconstruction operations, respectively. This adapter encourages frequency-aware modulation that preserves coarse spatial information while enhancing signal representation.

Our proposed Fourier Adapter, though structurally simple, leverages global frequency components to modulate features effectively. Compared to the spatial-domain ARC, its gated variant, and wavelet-domain alternatives, it consistently yields superior performance on image deraining presented in Table 11. This confirms the strong capability of frequency-domain adapters in capturing globally coherent structures essential for low-level vision adaptation.

## A.10 SPECTRAL ANALYSIS OF FRAIR

To better understand why FraIR yields stronger domain adaptation than prior Fourier-adapter designs, we analyze the learned spectral parameters across both attention and MLP branches. Unlike FouRA—which employs a fixed Fourier basis with a single scalar gate—FraIR learns a pair of structured spectral operators: a basis matrix $U \in \mathbb{R}^{D \times D'}$ and a frequency gate $\Lambda \in \mathbb{R}^{D'}$. Their interaction defines an effective spectral modulation matrix $U \odot \Lambda$, which directly controls how hidden representations are transformed in the frequency domain.

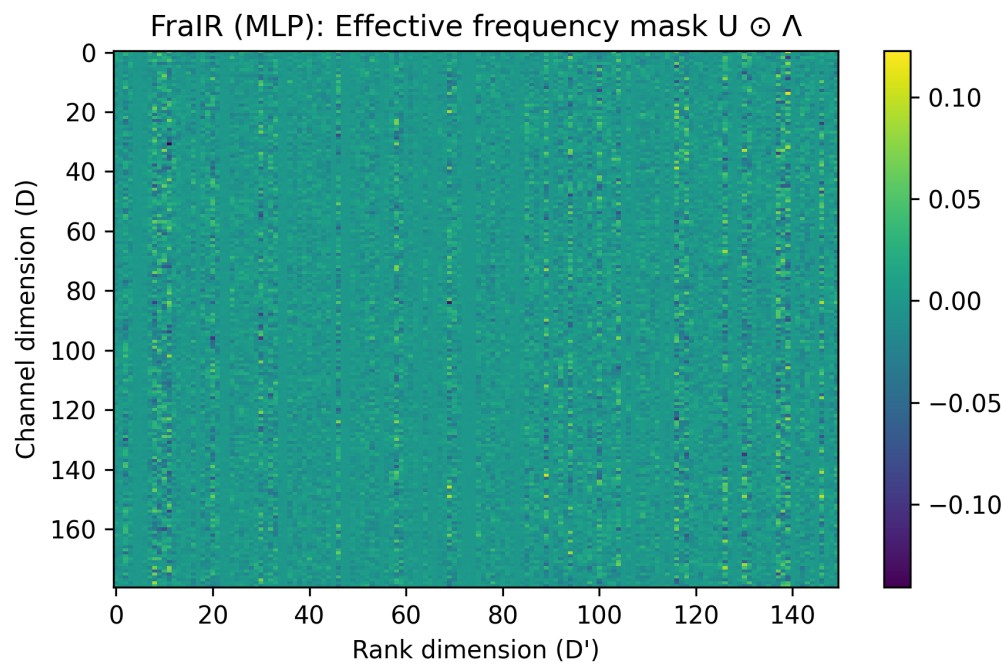

Figure 7: Learned frequency mask of FraIR (MLP branch). We visualize the effective spectral modulation matrix $U \odot \Lambda$, where $U \in \mathbb{R}^{D \times D'}$ is the learned spectral basis and $\Lambda \in \mathbb{R}^{D'}$ is the frequency gate. FraIR learns structured, non-uniform frequency responses across channels and ranks, unlike FouRA which applies a fixed linear projection.

**Learned spectral mask.** Figure 7 visualizes the effective spectral mask $U \odot \Lambda \in \mathbb{R}^{D \times D'}$ for the MLP branch on the denoising task. Rows correspond to feature channels and columns to low-rank spectral directions. The mask exhibits clear structured patterns: a small subset of spectral directions is consistently amplified (vertical bands), while channel-wise variability (horizontal structure) indicates that FraIR modulates frequencies differently across semantic dimensions. This behavior demonstrates that FraIR learns non-trivial, degradation-aware spectral operators, in stark contrast to FouRA's fixed basis and global scalar gate. The emerging sparsity and anisotropy show that FraIR identifies compact, task-relevant spectral modes rather than uniformly altering all frequencies.

**Blockwise spectral activity.** To further quantify how FraIR distributes spectral corrections throughout the network, we compute the Frobenius norm $\|U \odot \Lambda\|_F$ for each CSwin block (Figs. 8, 9). Both branches show a characteristic pattern: early blocks exhibit significantly higher spectral activity, while deeper blocks require weaker corrections. This aligns with the intuition that real-world degradations (noise, blur, compression) predominantly affect early low-level representations, whereas higher layers require only mild spectral refinement. Notably, the attention branch shows stronger overall modulation, suggesting that FraIR leverages frequency adaptation primarily within contextual aggregation pathways.

For clarity, in table 15 we explicitly detail how the proposed Fourier-based Recomposition Adapter (FRA) corresponds to its implementation in the transformer backbone.

## A.11 MORE VISUAL COMPARISONS

In this work, we conduct a comprehensive evaluation of various PETL methods across a broad range of image restoration tasks involving multiple training and testing datasets. For clarity, a detailed summary of the datasets used is provided in Table 14. In addition, visual comparisons are presented in Figure 10, where FraIR effectively removes complex degradations such as noise and rain while faithfully preserving fine details consistent with the ground truth.

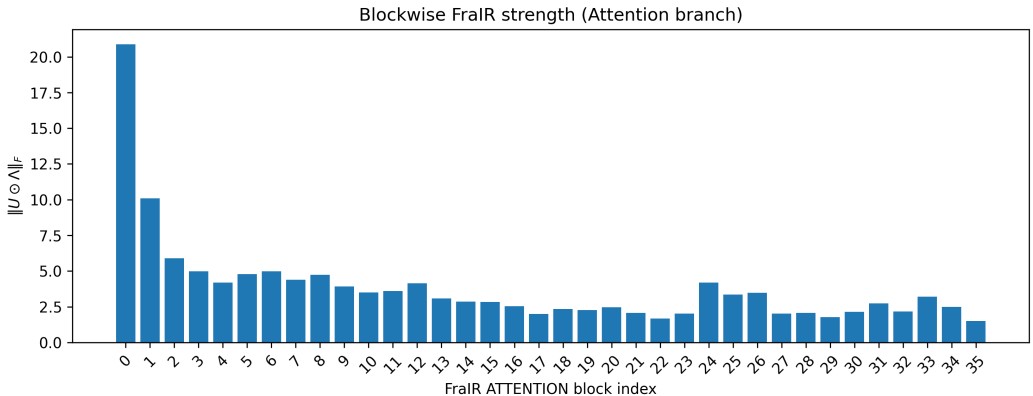

Figure 8: **Blockwise FraIR strength in the attention branch.** For each CSwin block, we compute the Frobenius norm $\|U \odot \Lambda\|_F$ to measure the magnitude of frequency-domain adaptation. FraIR is most active in early blocks, where real degradation effects are strongest, and gradually decreases in deeper layers.

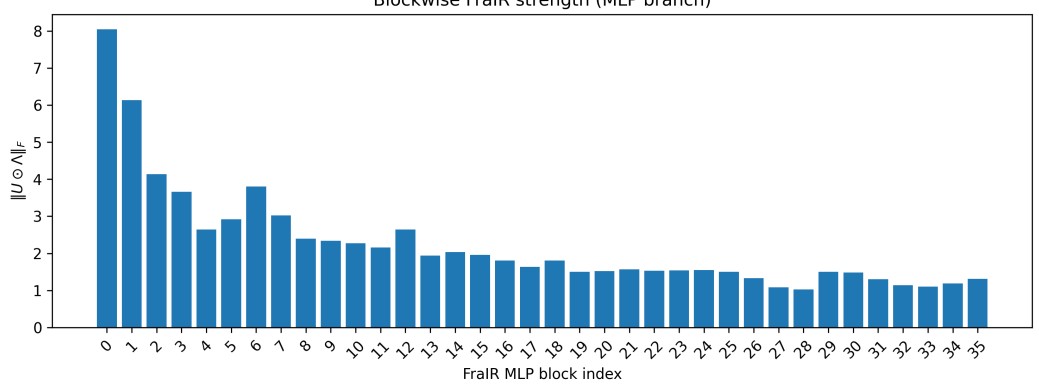

Figure 9: **Blockwise FraIR strength in the MLP branch.** Similar to the attention pathway, FraIR applies stronger spectral corrections in early layers and stabilizes in deeper blocks, indicating that low-level distortions dominate the domain gap.

Table 14: Dataset description for various image restoration tasks.

| Tasks | Type | Dataset | Num_samples |
|---|---|---|---|
| Denoise | train | BSD400 + WED | 400 + 4744 |
| | test | BSD68 + Urban100 | 68 + 100 |
| Derain | train | RainTrainL | 200 |
| | test | Rain100L | 100 |
| Super-Resolution and Hybrid Degradations | train | Div2K + Flickr2K | 800 + 2650 |
| | test | Set5 + Set14 + BSDS100 + Urban100 + Manga109 | 5 + 14 + 100 + 100 + 109 |
| Real-world Super-Resolution | train | DIV2K + RealSR + DRealSR | 800 + 400 + 840 |
| | test (patch) | DIV2K3000 + RealSR +DRealSR | 3000 + 100 + 93 |

| Mathematical Symbol | Meaning | FRA Implementation |
|---|---|---|
| $U \in \mathbb{R}^{D \times D'}$ | Projection basis | `u (attn, mlp)` |
| $\Lambda \in \mathbb{R}^{D \times D'}$ | Elementwise spectral scaling | `lambda_ (attn, mlp)` |
| $U \odot \Lambda$ | Scaled basis | `u * lambda_` |
| $\hat{Z} = \hat{X}(U \odot \Lambda)$ | Low-rank spectral transform | `x @ (u * lambda_)` |
| $\hat{Z}' = \hat{Z}U^{\top} + \mathbf{b}$ | Recomposition + bias | `adapted @ u.t() + bias` |
| $\mathbf{g} \in \mathbb{R}^{D}$ | Channel-wise gate | `gate_attn, gate_mlp` |
| $X_{\text{out}} = X + \mathbf{g} \odot \tilde{X}$ | Residual fusion | `x + gate * adapted` |

Table 15: Direct correspondence between the mathematical FRA formulation and its implementation.

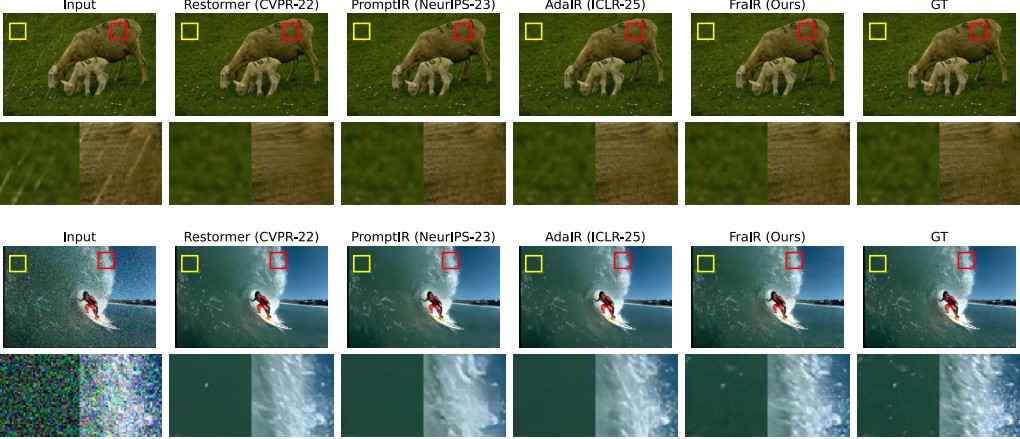

Figure 10: For qualitative results, we compare FraIR to Restormer Zamir et al. (2022), PromptIR Potlapalli et al. (2023) and AdaIR Cui et al. (2025) for different degradations. FraIR effectively removes rain streaks and noise injections while preserving image sharpness, achieving high-quality restoration.