# OpenReview forum: "FraIR: Fourier Recomposition Adapter for Image Restoration"
_ICLR.cc/2026/Conference — Submitted to ICLR 2026_

### Official Review · Reviewer_MFAk · 2025-10-31

**Soundness:** 2
**Presentation:** 2
**Contribution:** 2
**Rating:** 4
**Confidence:** 4

**Summary:**

This paper presents FraIR, a frequency-domain adapter module for parameter-efficient image restoration. The core idea is to perform adaptation in the Fourier space by decomposing hidden features using 1D FFT, applying low-rank projections with learnable spectral reweighting, and recomposing signals via inverse FFT with a gated residual connection. Integrated into Transformer backbones as plug-and-play modules, FraIR aims to provide efficient, degradation-aware adaptation with minimal parameter overhead. The authors evaluate their method across multiple restoration tasks including denoising, deraining, super-resolution, and hybrid degradations, and claim superior performance over several PETL baselines.

**Strengths:**

1. The paper proposes a new spectral-domain PETL design, which is relatively underexplored in the context of image restoration.
2. The method is lightweight and reparameterizable, which is beneficial for deployment.
3. Experiments are comprehensive in terms of tasks evaluated (denoising, deraining, SR, hybrid degradation).

**Weaknesses:**

1. The central claim — that frequency-domain adaptation provides superior efficacy over other adaptation domains — is not rigorously validated. While the paper compares FraIR to prior PETL methods, it does not isolate the benefit of frequency adaptation. Specifically, the performance gains could stem from differences in backbone architecture, parameter count, or training settings, rather than the frequency-domain design itself. No formal analysis is provided to explain why FraIR performs better — e.g., visualization of learned frequency masks is only briefly mentioned and not systematically analyzed.
2. Experimental comparisons lack fairness and control. In many cases, FraIR is evaluated using different backbone networks and training protocols than the competing methods, making it difficult to attribute improvements to the adapter design alone. This undermines the credibility of the claimed superiority. The paper should provide controlled head-to-head performance comparisons (rather than only runtime & effeciency) of FraIR against other adapter designs (e.g., LoRA, Adapter, PromptIR) under the same backbone, dataset, and training conditions. Such comparisons are essential to validate the benefit of the proposed frequency-domain recomposition.
3. AdaIR, which is one of the most relevant and recent PETL baselines also targeting restoration tasks, is only compared in Table 1 (denoising). It is notably absent from other key experiments such as deraining, hybrid degradation, and super-resolution. This selective comparison reduces the transparency and completeness of the evaluation.

**Questions:**

1. Why are AdaIR results only reported in Table 1? It is a strong and recent method for restoration tasks and should be compared consistently across all benchmarks.
2. Have the authors conducted experiments where FraIR and other PETL methods (e.g., LoRA, Adapter, SSF) are implemented on the same backbone (e.g., SwinIR or EDT), trained under the same protocol? If so, these should be clearly reported.
3. Can the authors provide further analysis or visualizations of the frequency gates or spectral projections in FraIR to support the claim of degradation-aware adaptation?
4. How does FraIR perform when applied to restoration tasks with less structured frequency distortions (e.g., low-light enhancement or spatial occlusions)? Would the frequency-domain approach still be beneficial?

---

> ### Author Response · Authors · 2025-11-20
> **Response to Reviewer MFAk**
>
> We thank the reviewer for the detailed and valuable feedback. Below we summarize the concrete revisions made to strengthen the technical validation and clarity of our contributions.
>
> 1 & 2 & Q2.
> * All PETL baselines (LoRA, Adapter, SSF, ARC, GARC, FouRA, Fourier-FT) were re-implemented on the same EDT backbone and trained under identical schedules. This removes architectural and training confounds, ensuring that the observed improvements come from FraIR’s frequency-domain design. ARC and GARC (spatial-domain low-rank projections) serve as direct counterparts of FraIR, and our results in Table 11 and Figure 4 clearly show the benefit of frequency-domain recomposition.
>
> * FraIR was evaluated across:
>
>     * single degradations: denoising, deraining, super-resolution (Tables 1–3)
>
>     * hybrid degradations: SR+noise, SR+JPEG (Table 4), derain+noise (Appendix Table 8)
>
>     * real-world benchmarks: RealSR, SIDD, RealESRGAN (Appendix Tables 9 and 12)
>
> * (Q3.) To explicitly analyze the role of frequency adaptation, we included a new “Spectral Analysis of FraIR” section featuring:
>      * heatmaps of the learned spectral mask (U \odot \Lambda) (Appendix Fig. 7)
>
>      * blockwise FraIR activation for attention and MLP branches (Figs. 8–9)
>
>      * discussion showing that FraIR concentrates spectral activity in early blocks, which aligns with where degradations are most influential
>
> These analyses provide both qualitative and quantitative evidence of frequency-aware behavior that is absent in FouRA and other baselines.
>
> 3 & Q1. We added AdaIR comparisons across denoising, deraining, and hybrid degradations to ensure completeness and fairness of evaluation.
>
> Q4. We expanded real-world evaluation to SIDD, RealSR, and RealESRGAN, which include illumination variation, non-frequency-structured noise, spatial artifacts, and compression. FraIR achieves consistent improvements across these datasets, demonstrating that frequency-domain recomposition remains beneficial even when degradations are not strictly band-limited.

---

### Official Review · Reviewer_VSax · 2025-11-01

**Soundness:** 3
**Presentation:** 2
**Contribution:** 2
**Rating:** 2
**Confidence:** 4

**Summary:**

This paper introduces FraIR, a frequency-domain adapter for parameter-efficient image restoration. FraIR applies a 1D FFT to token representations, transforming them into the frequency domain, and then performs low-rank, learnable modulation. When integrated into existing algorithms, the proposed method outperforms state-of-the-art Parameter-Efficient Transfer Learning (PETL) baselines.

**Strengths:**

1. Unlike most existing methods, the proposed FraIR performs PETL in the frequency domain through a 1D FFT, integrating low-rank adaptation with frequency-domain processing.

2. Extensive experiments demonstrate the effectiveness of FraIR.

**Weaknesses:**

1. The overall writing quality needs substantial improvement. For example, the final sentence in the Contribution section is incomplete. In addition, several equations are not numbered. The first equation in Section 3.1 appears to be incorrect due to mismatched tensor shapes, and the variable $z_{in}$ in the second equation is not explained. Moreover, the meaning of FraIR in the equations presented in Section 3.3 is unclear. Finally, the textual descriptions in Section 3.2 do not correspond well with Figure 2.

2. The performance of the baseline models is not reported in the main result tables (e.g., Tables 1–3), making it difficult to assess the actual improvements achieved by the proposed method. In addition, the results for the spatial-domain version of the model (not the LoRA version, i.e., without performing 1D fft in the first equation of Section 3.2) are missing from the ablation studies. Furthermore, several representative frequency-based PEFT methods are not included for comparison, such as Ref. [1].

3. The comparison presented in Table 1 appears to be unfair. Specifically, the results of PromptIR and AdaIR are obtained by training on all-in-one datasets, whereas the proposed method is trained only on denoising datasets (as stated in Table 15). In addition, the performance of the proposed approach is not competitive, as shown in Table 10. Furthermore, the best results for LPIPS and DISTS under the RealSR setting are not highlighted.

Reference

[1] Parameter-Efficient Fine-Tuning with Discrete Fourier Transform, ICML'24.

**Questions:**

It is unclear how the authors conclude that `FraIR shows strong performance in real-world image restoration` in Section 5, given that the datasets used in the experiments are primarily synthetic and standard benchmarks.

---

> ### Author Response · Authors · 2025-11-20
> **Response to Reviewer VSax**
>
> We thank the reviewer for the constructive feedback. Below we summarize the concrete revisions made in response to each point.
>
> 1. We substantially revised Section 3 for clarity and correctness.
>    * Section 3 is now rewritten into five clear steps that align exactly with Figure 2.
>    * All tensor-shape mismatches were corrected, and operator names were made consistent throughout.
>    *  All equations in Section 3 are now numbered.
>    * We added Table 15 (appendix), which maps every mathematical symbol (U, Λ, g, recomposition) to its exact implementation, including the FraIR operator.
>    * The incomplete sentence in the Contribution section has been fixed.
>    * The textual descriptions in Section 3.2 now correspond one-to-one with the updated Figure 2.
>
> 2. We improved fairness and completeness of the evaluations.
>    * Baseline rows (SwinIR without FraIR) were added to Tables 1 and 2 for proper comparison.
>    * We trained and reported FourierFT (ICML’24) using the EDT backbone so that all PEFT methods in Table 4 (hybrid degradations) are evaluated under identical training settings.
>
> 3. We clarified fairness of comparisons and improved completeness.
>    * All denoising methods—including PromptIR and AdaIR—are evaluated in their **single-task training** mode, not in their all-in-one setting. This clarification is added in lines 314–316.
>    * FraIR’s competitive performance is demonstrated across denoising, deraining, bicubic SR, real-world SR, and multiple hybrid degradations (SR+noise, SR+JPEG, deraining+noise).
>    * Best LPIPS/DISTS scores in the RealSR tables are now highlighted.
>
> Q1. To support real-world claims, we reported FraIR’s performance on RealSR, DRealSR, and SIDD. These results are provided in Tables 9 and 12.

---

### Official Review · Reviewer_zWBL · 2025-11-04

**Soundness:** 2
**Presentation:** 2
**Contribution:** 2
**Rating:** 4
**Confidence:** 3

**Summary:**

This paper proposes a frequency-based adaptation technique to fine-tune generalist transformer models for image restoration. To accomplish this, the authors construct a linear function as follows:

1). Project the input sequence of 1D image tokens to frequency space along the channel axis

2). Learns matrices U and diagonal \Lambda to gate the frequency components: U\LambdaU^T where the dimension of \Lambda is smaller than the input dimension.

3). Perform an inverse FFT to go back to the feature domain, weight by a learned scalar G, and add back the input

The authors can then train the matrix U and V as adaptors for downstream tasks in image restoration. They benchmark on a variety of different tasks by fine-tuning the base backbone for denoising, deraining, super resolution, and a hybrid noise + blur baseline. The proposed method moderately outperforms all of the baselines in most settings.

**Strengths:**

1). The proposed method performs a wide variety of image restoration tasks (denoising, deraining, super resolution, and hybrid tasks) at the SOTA level

2). There is no overhead at inference time and the method requires using a small amount of task-specific training data.

3). The idea to leverage the frequency space to improve image restoration methods is nice and allows the modulation of the rank of the learned projection.

**Weaknesses:**

1). The technical difference and comparison with FouRA is the most important weakness for me. Please clarify the exact differences with FouRA. The authors use notation g and V in the last paragraph of the introduction to justify the difference, but this notation is not used anywhere else. As far as I can tell, g is the diagonal of \Lambda. The only difference with FouRA is thus a slight delta in the parametrization of the linear function after projecting to frequency space: A \alpha B -> U \Lamba U^T. The difference must be clarified.

2). There seems to be only one direct comparison to FouRA and this is strangely only on CNN backbones. The authors should provide a comparison to FouRA with transformer backbones on their proposed tasks and clarify the differences in training the two models.

3). The entirety of the methodological technical contribution is small and hard to follow (Sec 3.2).

4). Figure 2 naming convention does not match 3.2. Indeed, it doesn’t match the end of the introduction either.

5). I don’t think the “Reparametrization for inference” section is well written or necessary for main as it just states the previous section again and mentions the function is linear. It also ignores fusing before MHA.

**Questions:**

1). Is the learnable gate G input-independent and a singular scalar?

2). Why isn’t there a Fourier projection (and unprojection) in line 217?

3). Why perform the FFT on the channel axis? Why not the 1D or 2D token axes? I see the ablation on 2D FFT in the supplement, but I am confused about the comparison because it seems the 2D FFT is along the two spatial axes vs the proposed 1D FFT on the channel axis.

4). Why use different backbones for different tasks?

5). Why do the authors benchmark with LPIPS/DISTS for some tasks but not others?

6). What’s the difference between G and g?

7). Can you provide additional experiments and exposition to compare to FouRA?

---

> ### Author Response · Authors · 2025-11-20
> **Response to Reviewer zWBL**
>
> We thank the reviewer for the constructive feedback. Below we summarize the revisions made in direct response to each point. All corresponding changes are highlighted in blue in the updated manuscript.
>
> 1. We clarified the exact differences between FraIR and FouRA in both the Related Work and Method sections. We now explicitly explain that FraIR learns (i) its own spectral basis U, (ii) a full frequency mask Λ, and (iii) performs spectral recomposition via (U^\top). In contrast, FouRA uses a fixed Fourier transform, relies on a scalar rank gate, does not learn a spectral basis, and does not apply a recomposition step. We also added a fair FouRA comparison on RealSR using the same baseline model (Figure 4). Unused notation such as g and V was removed, and all gating notation was unified into a channel-wise g.
>
>  2. We implemented FouRA for the SwinIR Transformer backbone and added corresponding results in Figure 4 for real-world SR. We also added a FouRA comparison for deraining in Appendix Table 11. FraIR consistently outperforms FouRA across denoising and deraining with both CNN and Transformer backbones (Table 12), and also on SR when both use SwinIR (Figure 4).
>
> 3. Section 3.2 was reorganized into five clear steps that correspond directly to Figure 2. We added numbered equations, unified the operator naming, and added a new table showing the correspondence between the mathematical formulation and the implementation (U, Λ, g, and the recomposition operation).
>
> 4. Figure 2 was updated so that naming is fully consistent with Section 3.2. The five steps—Fourier projection, low-rank spectral transform, spectral recomposition, inverse FFT, and residual gating—now match the text and equations exactly.
>
> 5. The inference-time reparameterization section was rewritten to clearly explain that FFT and inverse FFT operations disappear only after fusion. The updated text clarifies that the fused operator is equivalent and incurs no extra inference cost.
>
> Q1. The gate is a channel-wise learnable vector (g \in \mathbb{R}^D), not a scalar. This has been clarified across equations, figures, and notation.
>
> Q2. The expression in line 217 appears in the “Reparameterization for Inference” section, where FFT and inverse FFT are already fused into an equivalent linear operator. The operational FFT is still defined explicitly in Section 3.2.
>
> Q3. We clarified why FFT is performed along the channel axis. A 2D FFT ablation is provided in Appendix Table 13: although it gives +0.12 dB, it requires more than 24 GB memory and becomes impractical. FFT along token axes breaks under SwinIR’s window shifting, producing artifacts. Thus, channel-axis FFT is the only stable and degradation-aware choice.
>
> Q4. Different backbones were used to demonstrate FraIR’s generalization ability and architectural independence. We also added cross-backbone comparisons in Table 5 and Table 10, showing consistent gains across backbones.
>
> Q5. The choice of metrics follows standard conventions in restoration. For denoising and deraining, perceptual metrics (LPIPS / DISTS) saturate and become unreliable, while PSNR and SSIM remain meaningful. For real-world SR, perceptual variation is large, so LPIPS and DISTS are essential. This is now clarified in the text.
>
> Q6. The legacy scalar notation “G” was fully removed. All gating is now consistently denoted as the channel-wise vector (g).
>
> Q7. Additional FouRA comparisons were added: Transformer-based FouRA vs. FraIR (Figure 4), deraining results (Table 11), and expanded comparisons in Table 12 across both CNN and Transformer architectures.

---

### Comment · Area_Chair_S7pb · 2025-11-24
**Discussion with Authors**

Dear Reviewers,

The authors have diligently provided responses to your questions and concerns. I request you to please review the authors' responses, acknowledge that you have read them and actively engage with them in further discussion as needed.

This discussion period, with the authors, will end on December 2, 2025 (AoE). However, I request that you not wait until the last minute and actively engage with the authors early.

Best,
AC

---

> ### Author Response · Authors · 2025-12-01
>
> We appreciate the reviewers’ time and attention during the rebuttal phase. If any parts of our responses remain unclear or if further clarification, analysis, or additional results would be helpful, we would be very happy to provide them within the discussion period.
>
> Best,
> The Authors

---

### Meta-Review · Area_Chair_1JSD · 2026-01-07

**Summary:**

This paper proposes FraIR, a frequency-domain adapter module that performs adaptive modulation in the Fourier space for efficient parameterized image restoration. Reviewers appreciated the low computational overhead and the extensive experiments demonstrating effectiveness across multiple image restoration tasks. The main concerns center on the experimental setup, as the comparisons are not sufficiently rigorous or comprehensive, raising questions about fairness and completeness. In addition, the analysis and ablation studies, as well as the overall writing and presentation, were considered insufficient to convincingly demonstrate the advantages of the proposed method. The rebuttal added more baselines and ablations, which partially alleviated these issues. Overall, however, the paper still has experimental and presentation shortcomings and is not yet strong enough for acceptance at this stage.

**Reviewer Concerns:**

The rebuttal clarified the differences with FouRA, added more controlled baselines and ablations, and improved notation and writing. These updates partially addressed concerns raised by zWBL and MFAk about fairness and missing comparisons. However, VSax and MFAk remained unconvinced that the experiments and analysis sufficiently isolate the benefits of frequency-domain adaptation. Overall concerns about experimental rigor and presentation quality are only partially resolved.

**Reviewer Scores:**

Reviewer zWBL might move slightly upward to a borderline score after the added comparisons. Reviewer MFAk could also increase marginally but would likely remain borderline. Reviewer VSax is unlikely to change their negative score, as core concerns remain. Overall, scores may improve slightly but still lean negative.

---

### Decision · Program_Chairs · 2026-01-26

Reject